# FreqRISE: Explaining time series using frequency masking

Thea Brüsch[*,1,2], Kristoffer Knutsen Wickstrøm[4], Mikkel N. Schmidt[1,2], Tommy Sonne Alstrøm[1,2], and Robert Jenssen[1,4,5]

[1]Department of Applied Mathematics and Computer Science, Technical University of Denmark
[2]Department of Physics and Technology, UiT The Arctic University of Norway
[3]Pioneer Centre for AI, University of Copenhagen, Denmark
[4]Norwegian Computing Center, Oslo, Norway
{theb,mnsc,tsal}@dtu.dk
{kristoffer.k.wickstrom,robert.jenssen}@uit.no

## Abstract

Time series data is fundamentally important for many critical domains such as healthcare, finance, and climate, where explainable models are necessary for safe automated decision-making. To develop explainable artificial intelligence in these domains therefore implies explaining salient information in the time series. Current methods for obtaining saliency maps assumes localized information in the raw input space. In this paper, we argue that the salient information of a number of time series is more likely to be localized in the frequency domain. We propose FreqRISE, which uses masking-based methods to produce explanations in the frequency and time-frequency domain, and outperforms strong baselines across a number of tasks. The source code is available here: https://github.com/theabrusch/FreqRISE.

## 1 Introduction

With the increasing development of systems for automated decision-making based on time series in critical domains such as healthcare [1, 2], finance [3] and climate forecasting [4], the demand for safe and trustworthy machine learning models is ever rising. High accuracy and explainability are both necessary components in achieving safety and trustworthiness. Deep learning models have become a popular choice to obtain high accuracy for time series tasks [5], but due to their complex reasoning process they are difficult to interpret. Explainable artificial intelligence (XAI) aims to open up the black box [6].

Most advancements in the field of XAI have been made in explaining image data [7–9]. Images hold the appeal of being easily interpretable by humans, making it simpler to validate model decisions by visual explanations. As such, many XAI methods provide explanations in the form of a relevance map over the pixels in the raw image [7, 9]. Time series do not benefit from the same properties, since they are by nature more difficult to understand [10]. While many time series models may be trained on raw time series data, the information (and thus relevance) may likely be found in a latent feature domain such as the frequency domain [11]. This challenge has only been addressed in [12], where the authors use gradient-based methods to backpropagate the relevance into the frequency domain.

Masking-based methods are an important tool in XAI due to their high performance across a multitude of domains [9, 13, 14]. Masking-based approaches learn or estimate the relevance maps by iteratively applying binary masks to the input and measuring the resulting change in the output. Especially optimization-based masking approaches have shown great promise for time series data [13, 15, 16]. In optimization-based masking approaches, the relevance is learned by posing an objective that maximizes the number of masked out features in the input, while simultaneously minimizing the change in the model output [17]. The mask is optimized via gradient descent through the model and the mask generating function. However, these methods can be difficult to use due to hyper-parameter tuning. Additionally, they suffer from the significant drawback of requiring access to the model gradients.

Alternatively, model-agnostic methods require only access to the input and output of the network and are easily adaptable to a multitude of network architectures. Prominent model-agnostic methods are also based on masking and involve estimating the relevance via Monte Carlo sampling. This category includes RISE [9] and RELAX [18]. The sampling based methods have a few clear advantages: they do not require any complex optimization techniques, and we can directly constrain the sampling space to ensure compatibility with our data. While these methods have been successfully applied for time series [19, 20], no previous work in this category has focused on providing relevance maps in another domain than the input domain.

In this paper, we take inspiration from RISE and propose the first approach for masking-based XAI in the frequency domain of time series, FreqRISE.

---

*Corresponding Author.

Proceedings of the 6th Northern Lights Deep Learning Conference (NLDL), PMLR 265, 2025.

We:

- Propose FreqRISE, the first masking-based approach for providing relevance maps in the frequency and time-frequency domain.

- Provide a comprehensive evaluation of our proposed approach across two datasets and four tasks.

- Show that masking-based relevance maps in the frequency and time-frequency domain outperform competing methods across several metrics.

## 2  Masking based explanations

Here, we present a new method for transforming masks between domains (Section 2.1) and use this new method to create FreqRISE (Section 2.2), the, to our knowledge, first masking-based XAI method operating in the frequency domain of time series.

### 2.1  Masking in a dual domain

While many time series models are trained directly on the raw time series data, the time domain is often insufficient for providing complete explanations [11]. This is due to most XAI methods being built on the assumption that the explanations are *localized* and *sparse*. However, if two classes are characterized by e.g. their frequency content, this information is neither localized nor sparse in the time domain. We, therefore, propose transforming the signals to a domain where the information is assumed to be localized and sparse and to provide the explanations in this domain.

Our aim is to explain the black box model $f(\cdot)$ that outputs the class probabilities $\boldsymbol{y} \in \mathbb{R}^C$ from a time series input $\boldsymbol{X} \in \mathbb{R}^{V \times T}$, where $V$ is the number of input variables and $T$ is the length of the time series. Masking-based methods, such as RISE [9] and RELAX [18], use masks $\boldsymbol{M} \in \{0,1\}^{V \times T}$ sampled from distribution $\mathcal{D}$, to mask out features in the input space. Here the $n$ index identifies the $n$-th sampled mask. The masking is done through elementwise multiplication such that $\hat{\boldsymbol{X}}(\boldsymbol{M}) = \boldsymbol{X} \odot \boldsymbol{M}$. The resulting change in the model output can then be computed:

$$\hat{\boldsymbol{y}}(\boldsymbol{M}) = f(\hat{\boldsymbol{X}}(\boldsymbol{M})). \qquad (1)$$

The result is therefore a relevance map over the input features in the input domain.

Instead, assume an invertible mapping from the time domain, $T$, into the domain of interest, $S$, $g : \mathbf{X}^T \to \mathbf{X}^S$. We can formulate an alternative masking-based approach, where the masks are applied in the new domain after which the input is transformed back into the time domain. One example of such mapping is the discrete Fourier transform (DFT), which maps the signal into the frequency domain, $\mathbf{X}^S \in \mathbb{C}^{V \times F}$. We can then apply masks $\boldsymbol{M} \in \{0,1\}^{V \times F}$ in the frequency domain and obtain the masked input in the time domain via the inverse mapping:

$$\hat{\boldsymbol{X}}(\boldsymbol{M}) = g^{-1}(g(\boldsymbol{X}) \odot \boldsymbol{M}). \qquad (2)$$

We can then use (1) to obtain the changes in the model output resulting from the mask.

In this paper, we focus on the frequency domain and the time-frequency domain, obtained through the DFT and the short-time DFT (STDFT). However, the formulation can be extended to other invertible mappings.

### 2.2  FreqRISE

A prominent masking-based framework is RISE [9]. RISE assumes that the masks $\boldsymbol{M}$ are sampled from a distribution $\mathcal{D}$. RISE then estimates the relevance, $\boldsymbol{R}_c(\lambda)$, of class $c$ for a point in input space, $\lambda$, as the expected value of the class probability $\hat{y}_c(\boldsymbol{M})$ (obtained using (1)) under the distribution of $\boldsymbol{M}$ conditioned on $\boldsymbol{M}(\lambda) = 1$, i.e. that the point is observed. Here, according to our definition, $\lambda = (v, t)$ if defined in the time domain, $\lambda = (v, f)$ in the frequency domain, and $\lambda = (v, t, f)$ in the time-frequency domain. For a full derivation of the mathematical details, see the appendix.

For a high accuracy classifier, we expect $\hat{y}_c$ to be to be low when the most important features are removed and, reversely, to be high when important features are not masked out. In practice, we can estimate the relevance using Monte Carlo sampling. We produce $N$ masks and estimate the expected value as a weighted sum, normalized by the expectation over the masks [9]:

$$\hat{\boldsymbol{R}}_c = \frac{1}{N \cdot \mathbb{E}[\boldsymbol{M}]} \sum_{n=1}^{N} \hat{y}_c(\boldsymbol{M}_n) \cdot \boldsymbol{M}_n. \qquad (3)$$

Since the masks are applied in the dual domain, the resulting relevance map is a map over relevances in this dual domain. In this work, we deal with univariate time series, i.e. $V = 1$, however the methods can be extended to multivariate cases.

We combine the RISE framework with our proposed frequency and time-frequency masking and call our method FreqRISE. The FreqRISE framework is shown in Figure 1 when using STDFT as the invertible mapping.

## 3  Experimental setup

We conduct experiments on two datasets: the synthetic dataset presented in [12] and AudioMNIST [21]. Both datasets have been shown to have salient information in either the frequency or time-frequency domain [12].

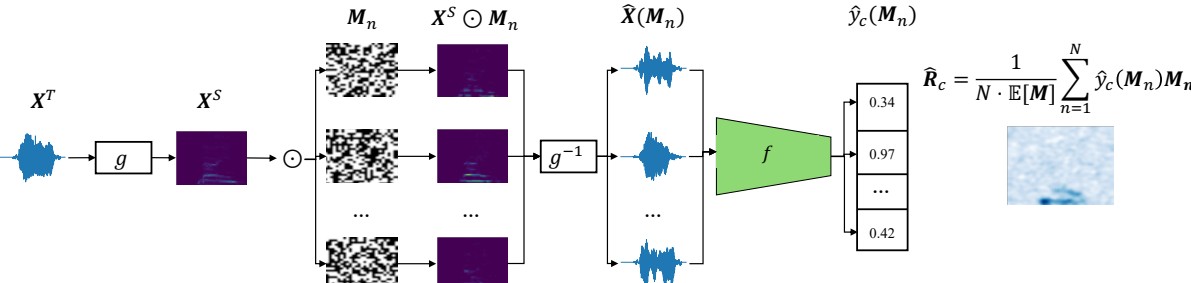

**Figure 1.** The FreqRISE framework shown when, as an example, using the Short-Time Discrete Fourier Transform (STDFT) as the invertible mapping, $g$. $g$ maps $\mathbf{X}^T$ into the domain of interest, $g : \mathbf{X}^T \to \mathbf{X}^S$. Here, we sample $N$ masks and produce $N$ masked versions of $\mathbf{X}^S$. Using the inverse of $g$, we map back to the time domain, obtain $\hat{y}_c$, and compute the relevance map as a weighted average over the masks.

## 3.1 Synthetic data

We use the same synthetic dataset as in [12]. The dataset is specifically designed to have the salient information in the frequency domain and therefore allows us to test the localization ability of the XAI methods. Each data point is created as a sum over $J$ sinusoids. $J$ is sampled uniformly from $\mathcal{U}(10, 50)$:

$$X_t = \sum_{j}^{J} a_j \sin\left(\frac{2\pi t}{T k_j} + \psi_j\right) + \epsilon. \qquad (4)$$

We set all $a_j = 1$, randomly sample the phase $\psi_j \sim \mathcal{U}(0, 2\pi)$, and add normally distributed noise $\epsilon \sim \mathcal{N}(0, \sigma)$. The length of the signals is set to $T = 2560$ and all frequency components are sampled as integer values within the range $k_j \sim \mathcal{U}\{1, 59\}$. We train the models to detect a combination of frequencies, $k^* \in \{5, 16, 32, 53\}$ from the time series signal. The classes are created from the powerset of $k^*$ and we therefore have 16 classes. Figure 2 shows a sample from the dataset. We train a 4-layer multilayer

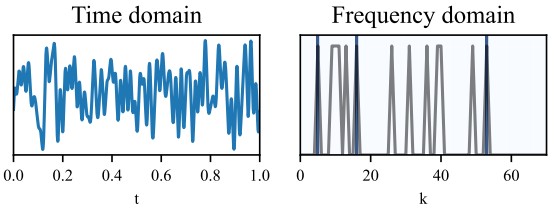

**Figure 2.** A sample from the synthetic dataset with salient features at $k = \{5, 16, 53\}$ marked in blue in the frequency domain.

perceptron (MLP) with a hidden size of 64 on two different versions of the raw time series: one with noise level $\sigma = 0.01$ and one with $\sigma = 0.8$. In both cases, we use $10^5$ samples for training. We test the models on a test set of size 1000 with no noise.

We use FreqRISE to compute relevance maps. We use the one-sided DFT to transform the signals to the frequency domain and sample binary masks, zeroing out single frequencies with $p = 0.5$ from a Bernoulli distribution. Similar to recent work [18], we use $N = 3000$ masks to obtain relevance maps.

## 3.2 AudioMNIST

AudioMNIST is a dataset consisting of 30,000 audio recordings of spoken digits (0-9) repeated 50 times for each of the 60 speakers (12 female/48 male) [21]. We follow [21] and downsample all signals to 8kHz and zero-pad all windows to 8000 samples.

We use the same 1D convolutional architecture as in [21] on the raw time series and train two versions: one for predicting the spoken digit and one to predict the gender of the speaker. Since the fundamental frequency is known to be discriminative for gender [22], we expect the salient information for the gender task to be localized in the frequency domain. The digit task, however, will likely be localized in both time and frequency, since we expect the numbers to be discriminated both through their formant information [23] (frequency domain) and the ordering of these in time (time domain).

We use FreqRISE to compute relevance maps in the frequency and time-frequency domains and standard RISE in the time domain. We use 1000 data points from the gender and digit test sets respectively for testing the explanation methods. All relevance maps for AudioMNIST are estimated using $N = 10,000$ sampled masks. All masks are sampled from a Bernoulli distribution with $p = 0.5$ on a lower dimensional grid and linearly interpolated to create smooth masks. When computing the (Freq)RISE relevance maps, we use the logits prior to the softmax activation, since these show higher sensitivity to changes in the input.

In the time-frequency domain, we use the one-sided STDFT with a Hanning window of size 455 and an overlap of 420 samples between subsequent windows following [21]. In the STDFT domain, we use binary grids of size $25 \times 25$. In the time and frequency domains, we use grids of size 200.

## 3.3 Baselines

As baselines, we use Integrated Gradients (IG) [24] and Layer-wise Relevance Propagation (LRP) [7]. When computing relevance maps in the frequency and time-frequency domains, we follow [12] and equip the models with virtual inspection layers, which map the input into the relevant domains. Due to relevance conservation, we are restricted to using rectangular, non-overlapping windows for the STDFT. We therefore use rectangular windows of size 455 with no overlap.

## 3.4 Quantitative evaluation

Due to the lack of ground truth explanations [25], quantitative evaluation for XAI is performed by measuring desirable properties [26]. Below we describe three such desirable properties that we use to quantify the quality of the relevance maps.

**Localization:** Localization scores are widely used to evaluate if explanations are co-located with a region of interest [13, 18]. For the synthetic data, we know the ground truth explanations and can therefore use localization metrics to evaluate the methods. We use the *relevance rank accuracy* [27]. Assuming a ground truth mask, $GT$, of size $K$ and a relevance map $\hat{R}$, we take the $K$ points with the highest relevance. We then count how many of these values coincide with the positions in the ground truth mask. Formally, for $P_{topK} = \{p_1, p_2, \ldots, p_K | \hat{R}_{p_1} > \hat{R}_{p_2} > \cdots > \hat{R}_{p_K}\}$, i.e. $p$ denotes the position, we compute:

$$\text{Relevance rank accuracy} = \frac{|P_{topK} \cap GT|}{|GT|}. \quad (5)$$

**Faithfulness:** Faithfulness measures to what degree an explanation follows the predictive behaviour of the model and is a widely used measure for quantifying quality of explanations [12, 16]. We follow prior works [12, 13] and compute faithfulness as follows. Given a relevance map, $\hat{R}$, we iteratively remove the $5\%, 10\%, \ldots, 95\%$ most important features by setting the signal value to 0. We then evaluate the model performance as the mean probability of the true class and plot it to produce deletion plots. Finally, we compute the area-under-the-curve (AUC) to produce our final faithfulness metric. A low AUC means that the explanation is faithful to the model.

**Complexity:** Finally, we evaluate the complexity of the explanation as an estimate for the informativeness [28]. The complexity is estimated as the Shannon entropy of the relevance maps.

## 4 Results

We present a comprehensive evaluation of FreqRISE using the experimental setup described in Section 3.

**Table 1.** Localization (L) and complexity (C) on the synthetic data.

|  | Low noise | | Noisy | |
| --- | --- | --- | --- | --- |
|  | L (↑) | C (↓) | L (↑) | C (↓) |
| IG | 99.1% | **1.10** | 52.4% | 1.96 |
| LRP | 99.1% | 1.11 | 63.8% | **1.34** |
| FreqRISE | **100.0%** | 7.09 | **100.0%** | 7.09 |

## 4.1 Synthetic data results

The two models trained on the synthetic dataset both achieve an accuracy of 100%. Table 1 shows the localization and complexity scores across the different methods for both the model trained on the low noise and the noisy datasets. The results on the low noise model show that all XAI methods perform approximately equal on the localization score, while IG and LRP yield relevance maps with substantially lower complexity scores compared to FreqRISE. However, when we move to the noisy model, the localization score of FreqRISE is unchanged, whereas both IG and LRP have much lower performance. The complexity of the IG and LRP relevance maps is slightly higher on the noisy model, while the complexity of the FreqRISE is unchanged. An example of the computed relevance maps is shown in the appendix.

## 4.2 AudioMNIST results

The model trained on the digit classification task achieves and accuracy of 96.9%, while the model trained on the gender tasks achieves an accuracy of 98.6%.

Following the procedure described in Section 3.4, we initially compute the faithfulness results. Figure 3 (top) shows the deletion plots for the digit classification task in the frequency and time-frequency domain. We have also included two additional baselines, namely randomly deleting features (Rand.) and deleting features based on their amplitude (Amp.). In both domains, the mean true class probability quickly drops. After dropping only 10% of the features, FreqRISE has a mean true class probability of 0.139 in the time-frequency domain and 0.248 in the frequency domain. After this, the value continues to drop with features being removed.

Figure 3 (bottom) shows the same results for the gender classification task. Here, the mean true class probability using FreqRISE is 0.435 after dropping only 5% of the features in the frequency domain, and the same value in the time-frequency domain is 0.445. The other methods drop to lower values in the time-frequency domain, indicating that FreqRISE struggles more to identify relevant features in the time-frequency domain.

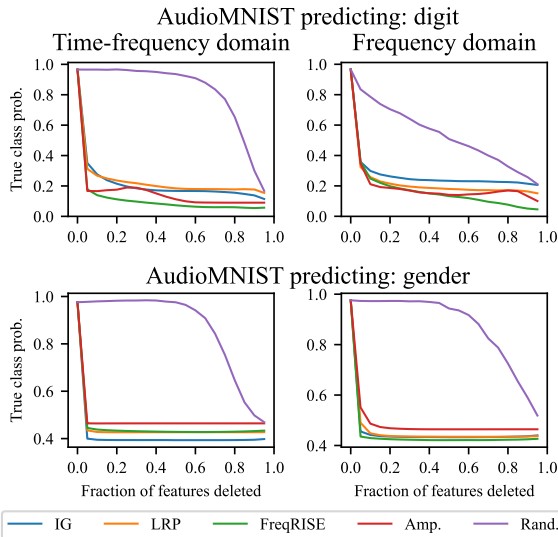

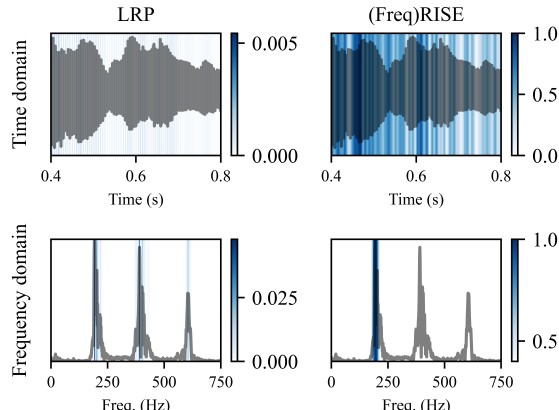

**Figure 3.** AudioMNIST: Deletion plots for both tasks in frequency and time-frequency domains. We delete features according to the importance and measure model outputs. FreqRISE outperforms the baselines in both domains on the digit task and the frequency domain on the gender task.

**Figure 4.** AudioMNIST: Gender task. We show relevance maps computed by LRP and FreqRISE in the time and frequency domain. The salient information is more localized in the frequency domain.

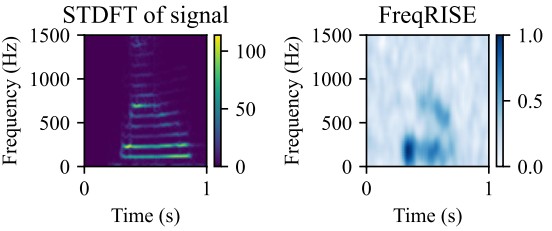

**Figure 5.** AudioMNIST: Digit task. STDFT of the signal on the left and the FreqRISE relevance map in the time-frequency domain on the right. The relevance map shows the benefit of having both the time and frequency axis, when the describing the digit data.

**Table 2.** AudioMNIST: Faithfulness (top) and Complexity (bottom) scores for explanations in the time (T), frequency (F) and time-frequency (TF) domains.

|  | Digit (↓) | | | Gender (↓) | | |
|---|---|---|---|---|---|---|
|  | T | F | TF | T | F | TF |
| IG | .183 | .252 | .197 | .470 | .428 | **.389** |
| LRP | **.168** | .205 | .214 | **.401** | .431 | .420 |
| (Freq)RISE | .233 | **.160** | **.104** | .419 | **.416** | .423 |

|  | Digit (↓) | | | Gender (↓) | | |
|---|---|---|---|---|---|---|
|  | T | F | TF | T | F | TF |
| IG | 6.93 | 6.41 | 5.26 | **6.57** | **4.74** | **4.04** |
| LRP | **6.88** | **5.84** | **4.67** | 6.78 | 5.16 | 4.16 |
| (Freq)RISE | 8.86 | 8.17 | 10.82 | 8.86 | 8.01 | 10.78 |

In Table 2 (top), the faithfulness scores are shown for both models in all three domains. We notice that in the time domain, RISE performs either worse or equivalently compared to LRP. However, when we move to the frequency domain, the faithfulness scores are lowest for FreqRISE in both cases. Finally, when we look at the time-frequency domain, the faithfulness score increases for FreqRISE on the gender task, while it decreases on the digit task. These results indicate that the information is sparser for the digit task in the time-frequency domain.

Table 2 (bottom) shows the complexity scores across all methods, domains and tasks. Again, it is clear that the complexity of the masking based methods is higher compared to the remaining methods.

Figure 4 (top) shows an example of relevance maps computed in the time domain on the gender classification task. Figure 4 (bottom) shows the same sample in the frequency domain. The sample is from a female speaker and the model correctly predicts the class. We see that LRP has a very sparse but scattered signal in the time domain, while RISE appears very noisy. Moving to the frequency domain, the relevance is much more localized, with FreqRISE putting most emphasis on the fundamental frequency, while LRP also focuses on the harmonics. The fundamental frequency is known to be discriminative of gender [22].

Figure 5 shows the relevance map computed using FreqRISE in the time-frequency domain for the digit task. The spoken and predicted digit is 9. The STDFT reveals that the signal starts around $t = 0.3s$, where the relevance map puts most of the relevance. However, around $t = 0.5s$, there are two distinct patterns emerging along the frequency axis. The fact that the method is allowed to distribute the relevance differently across frequencies depending on the time shows the benefit of having both the time and frequency component when computing relevance

maps for the digit task.

# 5   Discussion and conclusion

Time series data is inherently difficult to interpret, due to the complex patterns of the signals [10]. Therefore, providing relevance maps in a domain where information is sparser and more easily interpretable is desirable. We therefore proposed FreqRISE, which computes relevance maps in the frequency and time-frequency domains using model-agnostic masking methods. Vielhaben et al. [12] previously produced relevance maps in these domains using gradient based methods, which are model-dependent, limiting the usability in cases where model gradients are not available.

While LRP and IG are able to assign zero relevance, this is generally not the case for Freq(RISE). As a consequence, RISE and FreqRISE consistently yield high complexity scores. This is, however, not necessarily indicative of the perceptual quality of the relevance maps. The entropy could e.g. be reduced through post-processing of relevance maps. We have included preliminary results on a simple post-processing scheme in the appendix, showing that in most cases we can lower the complexity score to competitive levels while keeping a better faithfulness or localization score. Additionally, while complexity is a useful measure, it should always be considered in conjunction with localization and faithfulness scores.

We used a synthetic dataset with known salient features in the frequency domain to compute localization scores. In the low noise setting, FreqRISE gave slightly higher accuracy in identifying the correct frequency components. However, when testing in the high noise setting, FreqRISE outperformed the two baseline methods by a large margin (100% vs. 64%). These results indicate that the gradient-based methods (IG and LRP) both are susceptible to noise in the data, which results in a lower localization score in the noisy setting. Clearly, FreqRISE does not suffer from the same issues and instead has a stable, high performance independent of the noise level - likely because it is not dependent on the gradients of the model.

On AudioMNIST, we measured the faithfulness of all methods across three domains. RISE performed either slightly worse or similar to the baseline methods in the time domain. However, in the frequency domain FreqRISE gave the best faithfulness scores. In the time-frequency domain, FreqRISE gave the best performance on the digit task, but the worst performance on the gender task. The difference is likely to be found in the different data properties of the two tasks. This leads us to believe that FreqRISE is most suitable for providing explanations in domains where the information is assumed to be sparse. While the same information can theoretically be found in e.g. the time domain, the intricacy of the patterns would require more advanced masking schemes to identify the same patterns. For the gender task, the frequency domain is sufficient for describing the data since the fundamental frequency is known to be discriminative of gender [22]. Thus, when moving to the time-frequency domain, unnecessary complexity is added to the problem, resulting in a lower performance by FreqRISE. This leads us to believe that FreqRISE gives the best performance when computed in a domain where the salient information is sparse. As of now, no existing methods are able to estimate the "correct" domain in which to explain a model. Therefore, domain experts have to determine which domain is more useful for providing meaningful explanations. Future work should focus on developing metrics for measuring the most informative domains.

Finally, like many other explainabilty methods, FreqRISE requires choosing a suitable set of hyper-parameters. This includes choosing an appropriate size of the lower dimensional grid on which we sample the binary masks, as well as the number of masks to use. As of now, we have no principled way of choosing these hyper-parameters, although one solution could be to tune the faithfulness and complexity scores on a small validation set. Another, less systematic approach, simply requires the user to estimate the granularity of the problem and choose the grid size based on this. FreqRISE requires a rather large number of masks ($N = 10,000$) in order to produce high-quality relevance maps for AudioMNIST. This clearly increases the computational complexity compared to competing methods. Future work should investigate how to monitor convergence for potential early stopping.

In conclusion, FreqRISE gives stable, good performance in localization and faithfulness on two different datasets with a total of 4 different tasks. Additionally, unlike existing methods, FreqRISE is completely model-independent requiring no access to gradients or weights of the models being explained. As such, FreqRISE is an important next step in providing robust explanations in the frequency and time-frequency domain for time series data.

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

# A Appendix

## A.1 Details on the math behind FreqRISE

We closely follow the approach presented in [9], but extend the notation to include FreqRISE, where we allow for masking in dual domains. First, we reiterate the notation introduced in Section 2.1.

We consider a time series input $\boldsymbol{X} \in \mathbb{R}^{V \times T}$ transformed into some domain, $S$, through the invertible transform $g : \boldsymbol{X} \to \boldsymbol{X}^S$. Generally, $\boldsymbol{X}^S = \left\{ X^S \mid X^S : \Lambda \to \mathbb{R}^3 \right\}$ is of size $V \times T_S \times F$, i.e. $\Lambda = \{1, \ldots, V\} \times \{1, \ldots, T_S\} \times \{1, \ldots, F\}$. If $g$ is the Discrete Fourier transform $T_S = 1$, while if $g$ is the identity (i.e. we stay in the time domain) $F = 1$ and $T_S = T$. To ease notation, we will assume $V = 1$, but the math can easily be extended to the case where $V > 1$.

We then sample binary masks $\boldsymbol{M} : \Lambda \to \{0, 1\}$ from distribution $\mathcal{D}$. The perturbed input $\hat{\boldsymbol{X}}(\boldsymbol{M})$ is then obtained through elementwise multiplication with mask $\boldsymbol{M}$ in domain $S$ and subsequent inverse transformation through $g^{-1}$ back to the time domain:

$$\hat{\boldsymbol{X}}(\boldsymbol{M}) = g^{-1}\left(\boldsymbol{X}^S \odot \boldsymbol{M}\right). \tag{6}$$

We then obtain the outputs $\hat{y}_c(\boldsymbol{M})$ as the class prediction for class, $c$, using the black box model $f$:

$$\hat{y}_c(\boldsymbol{M}) = f\left(\hat{\boldsymbol{X}}(\boldsymbol{M})\right)_c. \tag{7}$$

Let us now define the importance, $\boldsymbol{R}_c(\lambda)$, of element $\lambda = (t, f)$ as the expected value over all possible masks $\boldsymbol{M}$, conditioned on $\lambda$ being observed, i.e. $\boldsymbol{M}(\lambda) = 1$:

$$\boldsymbol{R}_c(\lambda) = \mathbb{E}_{\boldsymbol{M}}\left[\hat{y}_c(\boldsymbol{M}) \mid \boldsymbol{M}(\lambda) = 1\right]. \tag{8}$$

Since $\boldsymbol{M}$ is binary, the expectation can be computed as a sum over masks, $\boldsymbol{M}'$:

$$\boldsymbol{R}_c(\lambda) = \sum_{\boldsymbol{M}'} \hat{y}_c(\boldsymbol{M}') P\left[\boldsymbol{M} = \boldsymbol{M}' \mid \boldsymbol{M}(\lambda) = 1\right]. \tag{9}$$

Using the definition of conditional distributions, this can be rewritten as:

$$\boldsymbol{R}_c(\lambda) = \frac{1}{P[\boldsymbol{M}(\lambda) = 1]} \sum_{\boldsymbol{M}'} \hat{y}_c(\boldsymbol{M}') P\left[\boldsymbol{M} = \boldsymbol{M}', \boldsymbol{M}(\lambda) = 1\right]. \tag{10}$$

Now, again since $\boldsymbol{M}(\lambda)$ is binary,

$$P\left[\boldsymbol{M} = \boldsymbol{M}', \boldsymbol{M}(\lambda) = 1\right] = \begin{cases} 0, & \text{if } \boldsymbol{M}'(\lambda) = 0. \\ P\left[\boldsymbol{M} = \boldsymbol{M}'\right], & \text{if } \boldsymbol{M}'(\lambda) = 1. \end{cases} \tag{11}$$
$$= \boldsymbol{M}'(\lambda) \cdot P\left[\boldsymbol{M} = \boldsymbol{M}'\right].$$

Additionally, $P[\boldsymbol{M}(\lambda) = 1] = \mathbb{E}[\boldsymbol{M}(\lambda)]$. Substituting this into (10):

$$\boldsymbol{R}_c(\lambda) = \frac{1}{\mathbb{E}[\boldsymbol{M}(\lambda)]} \sum_{\boldsymbol{M}'} \hat{y}_c(\boldsymbol{M}') \cdot \boldsymbol{M}'(\lambda) \cdot P\left[\boldsymbol{M} = \boldsymbol{M}'\right]. \tag{12}$$

We can now write this in matrix notation, such that the relevance can be computed for all elements, $\lambda$, simultaneously:

$$\boldsymbol{R}_c = \frac{1}{\mathbb{E}[\boldsymbol{M}]} \sum_{\boldsymbol{M}'} \hat{y}_c(\boldsymbol{M}') \cdot \boldsymbol{M}' \cdot P\left[\boldsymbol{M} = \boldsymbol{M}'\right]. \tag{13}$$

The relevance in (13) is then empirically estimated through Monte Carlo sampling. We sample $N$ masks, $\{\boldsymbol{M}_1, \ldots, \boldsymbol{M}_N\}$ according to $\mathcal{D}$ and normalize by the expectation of $\boldsymbol{M}$:

$$\boldsymbol{R}_c \approx \frac{1}{N \cdot \mathbb{E}[\boldsymbol{M}]} \sum_{n=1}^{N} \hat{y}_c(\boldsymbol{M}_n) \cdot \boldsymbol{M}_n, \tag{14}$$

finally arriving at Equation (3) in the main paper.

## A.2    Design choices for FreqRISE

As with other masking-based approaches, computing the relevance maps for FreqRISE involves choosing a number of hyper-parameters, i.e. the size of the grid in which to sample binary masks, the probability $p$ with which is bin is chosen, and the number of masks. As of now, there is no principled way to choose either and we have heuristically chosen the hyper-parameters based on qualitative assessment on a few validation samples. A more systematic approach, would be to tune the parameters on a validation set by choosing a few metrics, such as faithfulness and complexity.

For AudioMNIST, FreqRISE needs a large number of masks to converge ($N = 10,000$). We found that when using fewer masks, the method still finds the relevant bits, but sorting out the irrelevant bits of the signal requires more masks. Wickstrøm et al. [18] presented theoretical results on computing the number of masks and found that $N = 3,000$ should yield results with a low error. It would be interesting to investigate convergence properties for our datasets in light of the theoretical results.

Additionally, Mercier et al. [20] propose TimeREISE which uses a multiple mask sizes and sampling probabilities to compute the relevance maps for each time series. Finally, Cooper et al. [29] apply RISE to images and propose a hierarchical systematic mapping to reduce the number of masks. Both of these avenues could be interesting to explore when masking in the frequency domain.

## A.3    Post-processing of FreqRISE relevance maps

We do some preliminary experiments to reduce the complexity of the FreqRISE relevance maps through thresholding. Due to the properties of LRP and IG, both methods are able to assign zero relevance to specific points in the input space. This is not the case for FreqRISE, due to the weighted average with which the relevance maps are computed.

We post-process the FreqRISE results through thresholding. Given a relevance map $\boldsymbol{R}_c$, we produce a post-processed relevance map $\boldsymbol{R}_c^p$ by computing the $p$-th quantile, $q_p$ and setting all elements, $\lambda$, with value below $q_p$ to 0:

$$\boldsymbol{R}_c^p(\lambda) = \begin{cases} \boldsymbol{R}_c^p(\lambda), & \text{if } \boldsymbol{R}_c^p(\lambda) \geq q_p. \\ 0, & \text{otherwise.} \end{cases} \tag{15}$$

We compare the faithfulness and complexity trade-off for post-processed relevance maps of FreqRISE with different thresholds $p$. Additionally, IG and LRP are both able to assign negative relevance to specific points. However, often when visualizing the relevance maps, negative values will be set to zero [12]. In the main paper, we set negative values to zero when evaluating the complexity, but not when evaluating the faithfulness, giving both baseline methods an advantage in either case. Here, we therefore include LRP and IG results when setting all negative values to 0 (LRP pos. and IG pos.) and when considering positive and negative values (LRP and IG).

On the synthetic dataset, we set $p = 0.997$ and achieve a complexity score of 1.17 for both the low noise and high noise setting, while maintaining a localization score of 100% on both tasks.

All results are visualized in Figure A.1.

Ideally both faithfulness and complexity scores should be low. Thus, an ideal explanation method would give results in the lower, left quadrant of the plot. From Figure A.1 it is clear that with existing methods, faithfulness and complexity are typically trade-offs of each other.

For the digit task, FreqRISE consistently gives better faithfulness scores than the baselines independent of $p$. In the frequency domain, it is also clear that we are able to reduce complexity scores to a level closer to the best LRP complexity, while keeping a better faithfulness score (at $p = 0.8$). In the time-frequency domain, we are able to lower the complexity, although not to a competitive level. However, here the gap in faithfulness between FreqRISE and competing methods is quite substantial, indicating that maybe a higher complexity is needed to properly describe the data. Additionally, it is worth mentioning that for both LRP and IG using only the positive values does not influence the faithfulness score much, but it does lower the complexity.

For the gender task in the frequency domain, thresholds up to $p = 0.6$ still gives better faithfulness scores than the full LRP and IG relevance maps. Additionally, the complexity score of FreqRISE is comparable or lower compared to both IG and LRP. Using the non-thresholded FreqRISE relevance map gives better faithfulness and comparable complexity to full LRP and IG relevance maps. If we use only the positive values of LRP and IG, the faithfulness score is worse than FreqRISE at all thresholds, but the complexity is lower. In the time-frequency domain FreqRISE generally performs worse on both faithfulness and complexity. This is discussed in the main paper.

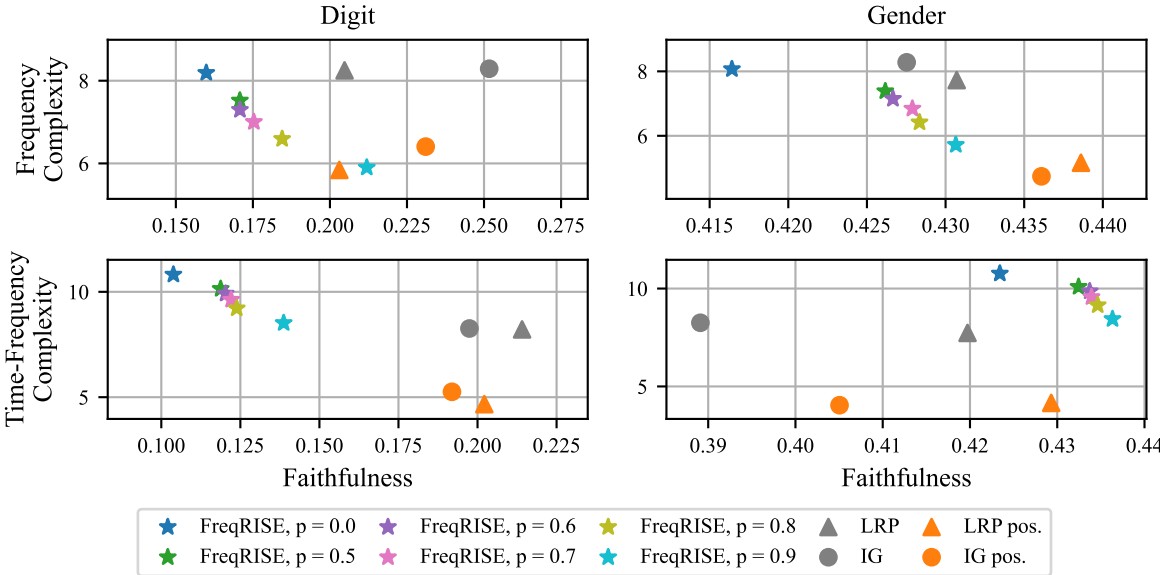

**Figure A.1.** Complexity vs. faithfulness for different post-processing of both FreqRISE, LRP and IG relevance maps.

Generally, we see that with simple thresholding of the FreqRISE relevance maps, we are able to reduce complexity of FreqRISE relevance maps while keeping competitive faithfulness scores.

## A.4 Additional relevance maps on AudioMNIST

Additional relevance maps in all domains on the digit and gender task for AudioMNIST.

Figure A.2 shows the same example as Figure 5 in the paper, but also in time and frequency. Figure A.2(a) shows that LRP seems to highlight most of the STDFT as important, seemingly ranked by the amplitude of the actual signal. This is not very useful for explanations. The time domain plots show that the time relevance of LRP seems to be centered around the middle of the signal. While for RISE, the time relevance is still scattered. In the frequency domain, the information seems more localized for both LRP and FreqRISE.

Figure A.3 shows an example on the digit task with the digit 0 being spoken. Here, we again see a distinct pattern in time and frequency for FreqRISE, while LRP seemingly focuses mostly on the lowest frequency component. In the time domain, the relevance is again scattered for both methods. In the frequency domain, the relevance is mostly localized on two frequency components for FreqRISE.

Figure A.4 shows the same example as Figure 4 in the main paper, but also in the STDFT domain. In the STDFT domain, we see that FreqRISE struggles to identify relevant information, whereas LRP distributes relevance on the three lowest frequency components. This indicates that the frequency domain is sufficient for explaining the gender task.

Figure A.5 shows an example on the gender task for a male speaker. In the STDFT domain, FreqRISE seemingly just highlights most of the signal, while LRP again puts emphasize on the three lowest frequency components. In the time domain, again information is scattered in time. Finally, in the frequency domain, FreqRISE puts most relevance on the fundamental frequency, while LRP places most relevance on the second harmonic of the fundamental frequency.

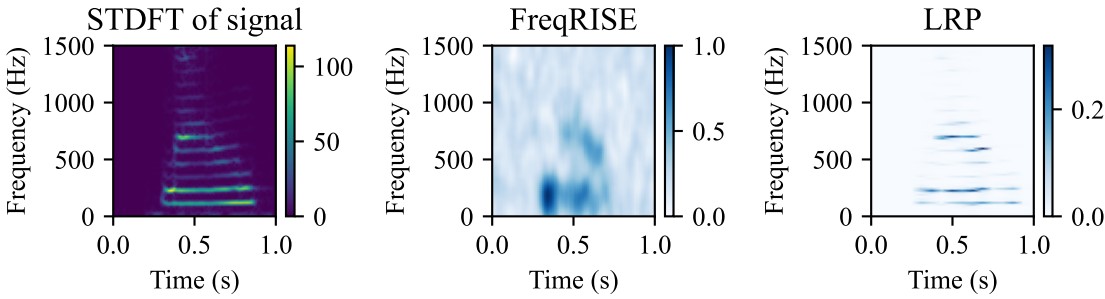

(a) Relevance maps in the STDFT domain.

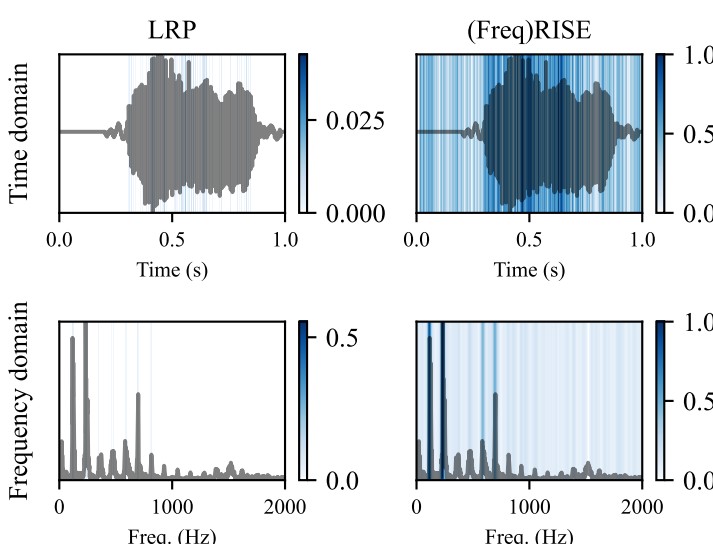

(b) Relevance maps in time and frequency domains.

**Figure A.2.** AudioMNIST: Digit task. Relevance maps in all three domains for a spoken digit 9 computed using LRP and FreqRISE.

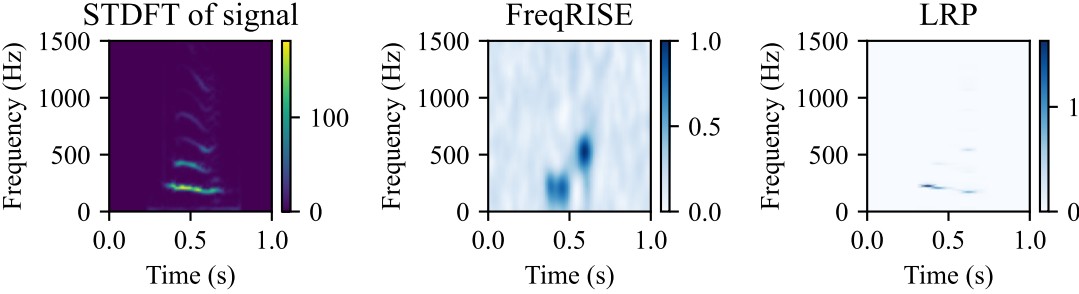

(a) Relevance maps in the STDFT domain.

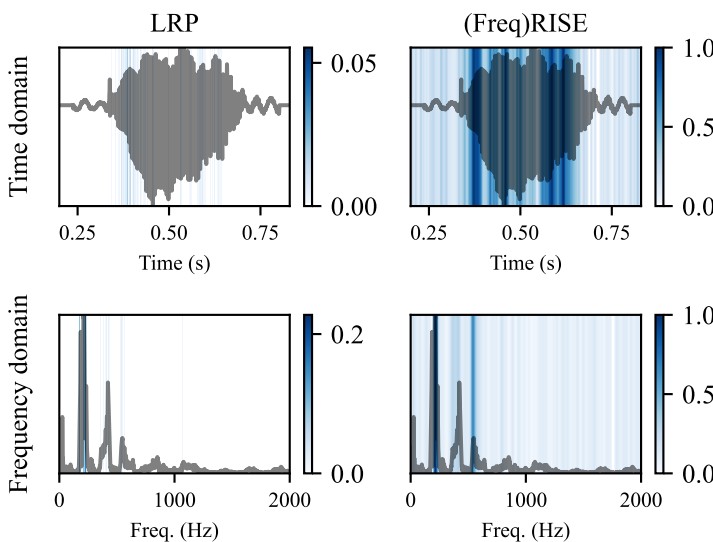

(b) Relevance maps in time and frequency domains.

**Figure A.3.** AudioMNIST: Digit task. Relevance maps in all three domains for a spoken digit 0 computed using LRP and FreqRISE.

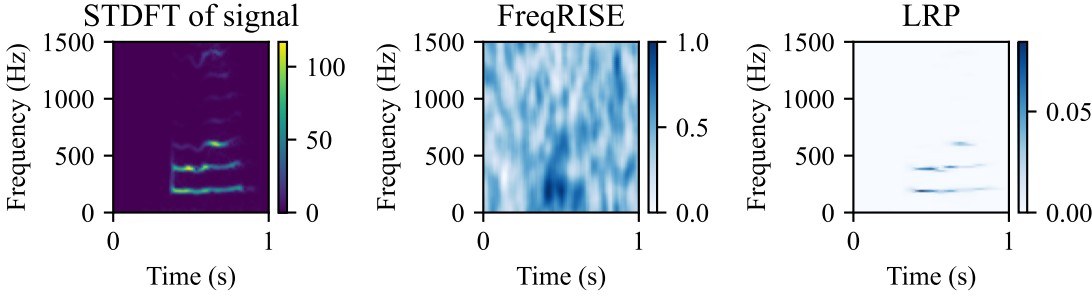

**(a)** Relevance maps in the STDFT domain.

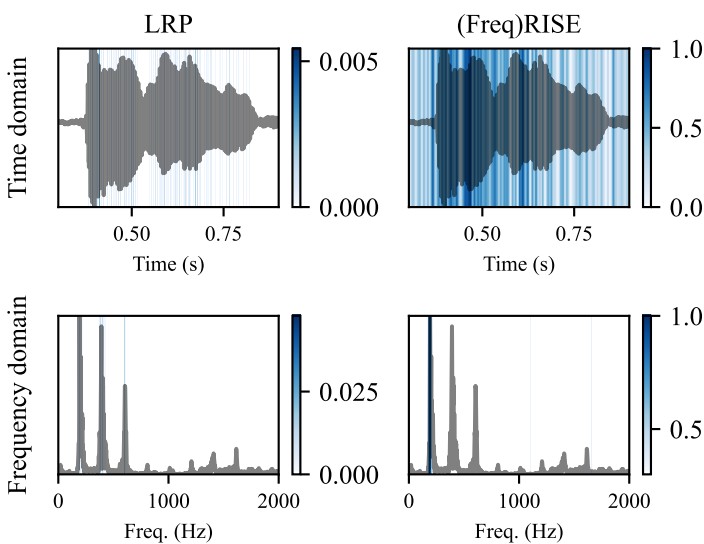

**(b)** Relevance maps in time and frequency domains.

**Figure A.4.** AudioMNIST: Gender task. Relevance maps in all three domains for a female speaker computed using LRP and FreqRISE.

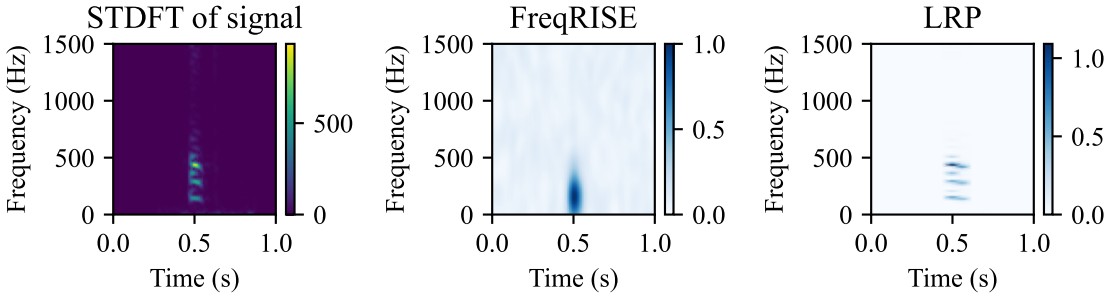

**(a)** Relevance maps in the STDFT domain.

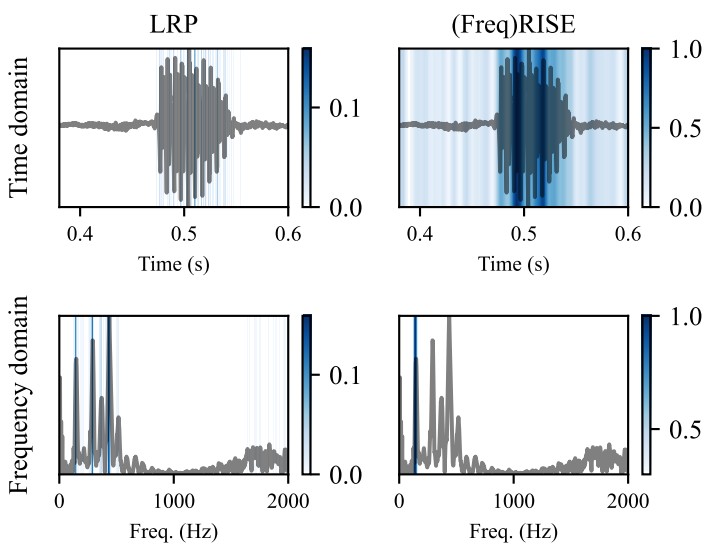

**(b)** Relevance maps in time and frequency domains.

**Figure A.5.** AudioMNIST: Gender task. Relevance maps in all three domains for a male speaker computed using LRP and FreqRISE.

## A.5   Results on synthetic data

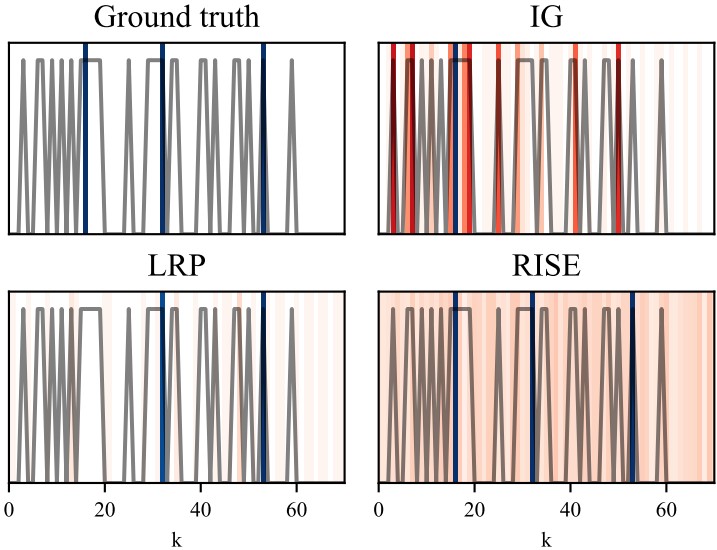

**Figure A.6.** Synthetic data: Relevance maps from the noisy model using each method along with the ground truth for a sample in dataset.

