# OpenReview forum: "FreqRISE: Explaining time series using frequency masking"
_NLDL.org/2025/Conference — NLDL 2025 Poster_

### Official Review · Reviewer_fUhF · 2024-10-04
**Introduction of a new explanation method for time series data**

**Confidence:** 4

**Summary:**

This paper builds upon existing explanation methods for time series data which utilize masking (RISE) and introduces the idea of applying masking to the frequency domain rather than input space. The outputs are then transformed back to the input space to obtain masks in the time domain, allowing explanations to be presented additionally in the time-frequency domain. The authors demonstrate the new technique, FreqRISE, in both an established synthetic dataset and the AudioMNIST dataset.

**Strengths:**

The paper is well organized and written, helping to make it accessible to a wide audience. Masking in the frequency domain is both interesting and appears to provide utility. Demonstrating performance in multiple ways on several datasets is useful.

**Weaknesses:**

There are some parts requiring further clarity both to the reviewer and future readers:
- In the Intro it is stated that there is a focus on imaging applications of XAI. It is worth adding other frequent applications of them eg on tabular and text data
- Some justification for the statement "We expect the salient information for the gender and digit task to be localized in the frequency and time-frequency domain, respectively." could be useful
- Clarification for why faithfulness was not computed and included in Table 1
- Please provide guidance on choosing parameters for FreqRISE, such as the number of masks. It is not immediately apparent why ~10x more are applied in the AudioMNIST task
- Some additional sample outputs for other classes (eg male speaker or other digits) would be interesting, in the appendices if required for the sake of space is fine

Fig 1
- The figure itself could do with some improvements to provide clarity eg at least some arrows etc to show the data flow, labelling of all parts (eg ŷ)
- More description of the pipeline is required in the caption. Currently it cannot easily be interpreted without first reading the text. For example, the meaning of STDFT must be included and additionally I suggest a brief description of the key stages of processing

Some minor typographical errors including:
- Line 102 misspelling "built"
- Line 227 "Freq" should presumably not be in brackets as currently appears

**Justification:**

The work is interesting and well written. The weaknesses are relatively minor and resolvable so on balance I recommend acceptance.

---

> ### Author Rebuttal · Authors · 2024-10-21
>
> We are happy to hear that the reviewer finds our paper well-written and our method interesting. We would also like to thank the reviewer for the helpful comments which will improve the paper. Please notice that we have uploaded a revised version of the paper as well.
>
> # General reply
> We include a general reply here, since more reviewers asked about the complexity scores.
> ## Updated faithfulness scores
> First of all, we want to mention that we found a small bug in how we had computed the faithfulness scores. We have updated the tables accordingly and uploaded a revised version of the paper. The absolute values have changed slightly, but none of them change any of the conclusions of the paper.
> ## Complexity
> The reviewers expressed concerns regarding the complexity of the relevance maps produced by FreqRISE. The reason for this is that, contrary to LRP and IG, FreqRISE is not able to assign 0 relevance to data points. This means that the baseline relevance will fluctuate on lower values, while FreqRISE will still assign high relevance to relevant points. We briefly mention that this might be solved through post processing. We did some preliminary experiments, where we set all values below a certain ratio of the maximum of the relevance map to zero.
>
> *Synthetic dataset*
> On the synthetic dataset we are able to achieve **complexities of 0.65 and 0.64** in the low and high noise settings respectively, while still achieving a **localization score of 100%** on both tasks.
>
> *AudioMNIST*
> Below, we show the complexity (C) and faithfulness (F) results that we are able to obtain on AudioMNIST (prior and post thresholding).
>
> | Task   | Domain         | C. prior | C. post | F. prior | F. post |
> | ------ | -------------- | -------- | ------- | -------- | ------- |
> | Digit  | Frequency      | 8.17     | 7.01    | .160     | .182    |
> | Digit  | Time-frequency | 10.82    | 8.44    | .104     | .134    |
> | Gender | Frequency      | 8.01     | 7.33    | .416     | .417    |
> | Gender | Time-frequency | 10.78    | 8.69    | .423     | .422    |
>
> The results clearly indicate that we can achieve lower complexity through thresholding, while keeping competitive faithfulness scores. We also want to emphasize that here, the threshold is chosen across the dataset, while a smarter thresholding may improve results further.
> Finally, while complexity is a good measure for the quality of the relevance map it is not in itself a complete picture. As such, low complexity with correspondingly low faithfulness or localization ability is not useful.
> We will extend the existing discussion in lines 383-389 to include the relevant points raised here. We will also include the preliminary post processing scores in the appendix.
>
> # Specific rebuttal
> - We agree that there are other active research areas within explainability. We will add [1] and [2] as references to tabular and text data applications as well.
> - Thank you for pointing this out. This statement is indeed a bit vague. We will update the sentence to:
> > “Since the fundamental frequency is known to be discriminative for gender [3], we expect the salient information for the gender task to be localized in the frequency domain. The digit task, however, will likely be localized in both time and frequency, since we expect the numbers to be discriminated both through their formant information [4] (frequency domain) and the ordering of these in time (time domain).”
> - Since we can compute the localization scores directly on the synthetic data, we see this score as sufficient for describing the method on the synthetic dataset (this adheres to the practice in e.g. [5]). Additionally, the salient information is localized on only 0.4% of the data (for the class with the most salient information), which makes it less meaningful to progressively remove features.
> - Thank you for raising this point, as this is definitely something that could do with some more discussion in the paper. Unfortunately, most explainability methods require tuning of hyperparameters (for LRP you have to choose the rules to use for backpropagation e.g. [6]). Regarding the number of masks, FreqRISE typically picks up on the relevant parts very early, while it usually takes more samples to remove the non-relevant parts. We have briefly discussed the selection of hyperparameters in the appendix, but will include these points in the discussion. Future works should investigate how to properly investigate the convergence to allow for quicker computation of explanations.
> - Thank you for the suggestion! We have included more examples in the appendix.
>
> Figure 1
> - Thank you for the useful suggestions! We have added more descriptors to the figure itself (including ŷ and arrows). Finally, we have also updated the Figure description to be more descriptive. Please see the revised version for the new figure.
>
> [1] Danilevsky, M., Dhanorkar, S., Li, Y., Popa, L., Qian, K., & Xu, A. (2021). Explainability for Natural Language Processing. Proceedings of the Acm Sigkdd International Conference on Knowledge Discovery and Data Mining, 4033–4034. https://doi.org/10.1145/3447548.3470808 \
> [2] Di Martino, F., & Delmastro, F. (2023). Explainable AI for clinical and remote health applications: a survey on tabular and time series data. Artificial Intelligence Review, 56(6), 5261–5315. https://doi.org/10.1007/s10462-022-10304-3 \
> [3] Calvert, D. R. (1988). CLINICAL MEASUREMENT OF SPEECH AND VOICE. Laryngoscope, 98(8), 905. https://doi.org/10.1288/00005537-198808000-00028 \
> [4] Brend, R. M. (1990). A PRACTICAL INTRODUCTION TO PHONETICS. Studies in Second Language Acquisition, 12(3), 352–353. \
> [5] Vielhaben, J., Lapuschkin, S., Montavon, G., & Samek, W. (2024). Explainable AI for time series via Virtual Inspection Layers. Pattern Recognition, 150, 110309. https://doi.org/10.1016/j.patcog.2024.110309 \
> [6] Montavon, G., Binder, A., Lapuschkin, S., Samek, W., & Müller, K. R. (2019). Layer-Wise Relevance Propagation: An Overview. Lecture Notes in Computer Science (Including Subseries Lecture Notes in Artificial Intelligence and Lecture Notes in Bioinformatics), 11700, 193–209. https://doi.org/10.1007/978-3-030-28954-6_10

---

### Official Review · Reviewer_Bf1T · 2024-10-07
**Promising concept but critical gaps in analysis and discussion**

**Confidence:** 4

**Summary:**

The authors introduce FreqRISE, an extension of the RISE explainability method applied to time series data by transforming it into the frequency and time-frequency domains. FreqRISE outperforms baseline methods in identifying relevant features, particularly in noisy environments and tasks where information is sparse. However, it consistently produces high complexity explanations, which may hinder interpretability and raises concerns about the correctness of the relevance maps, as they may include irrelevant information.

**Strengths:**

- **Core idea and novelty**: The paper introduces a novel application of the RISE explainability method to time series data by operating in the frequency and time-frequency domains, which is underexplored. This is a promising approach.
- **Performance in noisy environments**: FreqRISE demonstrates strong performance in noisy settings, in the context of the chosen synthetic dataset. This is an important result, as many real-world applications involve noisy time series data.
- **Clear introduction and methodology**: The paper does a good job of explaining the motivation behind the work and the FreqRISE method. The introduction is well-written, and the explanation of how the method works is detailed and easy to follow. The reasoning behind focusing on the frequency domain for time series analysis is clearly laid out, which helps build a solid foundation for the study.
- **Frequency domain advantage**: The method performs well on the AudioMNIST dataset when explanations are generated in the frequency domain. This indicates that FreqRISE is effective in identifying salient frequency components in this context.

**Weaknesses:**

- **Insufficient discussion of performance results**
    - In the synthetic dataset’s low-noise case FreqRISE performs the same as baseline methods, yet there is little discussion of what this means. Similarly, the fact that FreqRISE significantly outperforms in high-noise settings but not in low-noise ones raises questions about the method's behaviour and its reliance on noise levels. The implications of these results need to be explored further.
    - In the AudioMNIST task FreqRISE performs better in the frequency domain, but not in the time-frequency domain. The paper does not provide sufficient explanation of why FreqRISE’s performance varies between these domains. The authors should address what this discrepancy implies for different types of tasks and whether FreqRISE is generally more suited to purely frequency-based data.
- **High complexity of explanations**: FreqRISE consistently produces explanations with high complexity scores. This complexity can make the relevance maps difficult to interpret, limiting their practical usefulness. The paper doesn’t adequately discuss the consequences of this high complexity. The authors should explore whether the added accuracy is worth the trade-off in interpretability and whether post-processing could reduce this complexity.
- **Unclear argument for superiority**: The high complexity and mixed results make it difficult to conclude that FreqRISE is a superior method to the baselines. The lack of a detailed discussion around these limitations weakens the paper's overall argument. Without addressing how FreqRISE’s complex explanations might reduce its practical value, the claim of superiority over baseline methods is questionable.
- **Minor issues**: There are minor issues such as typos (line 102, “build”; line 177, “allows to”) and insufficient detail in the figure captions (figs. 3, 4, 5). Standalone figure descriptions would improve the clarity of the results and help readers better understand the key findings.

**Final Rebuttal Confidence:**

3

**Final Rebuttal Justification:**

I recommend accepting this paper, as it introduces a valuable approach to XAI for time series in the frequency domain. FreqRISE offers advantages in noisy data environments, demonstrating potential in challenging applications. While high complexity in relevance maps remains an issue, especially in low-noise contexts, the preliminary post-processing solutions are encouraging steps towards improved interpretability. Additionally, while guidance for selecting between frequency and time-frequency domains is only preliminary, this paper does provide a good basis for further development. Overall, this work is a promising addition to time series explainability and lays groundwork for future research in frequency-based interpretability.

**Justification:**

The paper introduces a promising method for explainable AI in time series data, but its contributions are unclear and unconvincing. Key findings lack in-depth analysis, particularly regarding the variation in performance for different tasks. Additionally, the method produces explanations with high complexity, which undermines interpretability. These shortcomings indicate that the contributions of the work are unclear and insufficiently supported.

---

> ### Author Rebuttal · Authors · 2024-10-22
>
> We are grateful that the reviewer finds our work promising and novel and the paper well written. Additionally, we would like to thank the reviewer for their comments and questions which will help improve the paper. Please notice that we have uploaded a revised version as well.
>
> # General reply
> We include a general reply here, since more reviewers asked about the complexity scores.
> ## Updated faithfulness scores
> First of all, we want to mention that we found a small bug in how we had computed the faithfulness scores. We have updated the tables accordingly and uploaded a revised version of the paper. The absolute values have changed slightly, but none of them change any of the conclusions of the paper.
> ## Complexity
> The reviewers expressed concerns regarding the complexity of the relevance maps produced by FreqRISE. The reason for this is that, contrary to LRP and IG, FreqRISE is not able to assign 0 relevance to data points. This means that the baseline relevance will fluctuate on lower values, while FreqRISE will still assign high relevance to relevant points. We briefly mention that this might be solved through post processing. We did some preliminary experiments, where we set all values below a certain ratio of the maximum of the relevance map to zero.
>
> *Synthetic dataset*
> On the synthetic dataset we are able to achieve **complexities of 0.65 and 0.64** in the low and high noise settings respectively, while still achieving a **localization score of 100%** on both tasks.
>
> *AudioMNIST*
> Below, we show the complexity (C) and faithfulness (F) results that we are able to obtain on AudioMNIST (prior and post thresholding).
>
> | Task   | Domain         | C. prior | C. post | F. prior | F. post |
> | ------ | -------------- | -------- | ------- | -------- | ------- |
> | Digit  | Frequency      | 8.17     | 7.01    | .160     | .182    |
> | Digit  | Time-frequency | 10.82    | 8.44    | .104     | .134    |
> | Gender | Frequency      | 8.01     | 7.33    | .416     | .417    |
> | Gender | Time-frequency | 10.78    | 8.69    | .423     | .422    |
>
> The results clearly indicate that we can achieve lower complexity through thresholding, while keeping competitive faithfulness scores. We also want to emphasize that here, the threshold is chosen across the dataset, while a smarter thresholding may improve results further.
> Finally, while complexity is a good measure for the quality of the relevance map it is not in itself a complete picture. As such, low complexity with correspondingly low faithfulness or localization ability is not useful.
> We will extend the existing discussion in lines 383-389 to include the relevant points raised here. We will also include the preliminary post processing scores in the appendix.
>
> # Specific rebuttal
> ## Performance results
> - Thank you for your questions. In the low noise setting, all methods achieve very high localization scores (approximately 100%). It is therefore not really possible for FreqRISE to outperform the competing methods. However, when we move to the high-noise setting FreqRISE has a stable high performance, while the performance of the baselines drops. This leads us to believe that the gradient-based methods (IG and LRP) both are susceptible to noise in the model and that this noise has a big influence on the gradients - and thus on the quality of the relevance maps. Clearly, FreqRISE does not suffer from the same issues and instead has a stable, high performance independent of the noise level. Likely because it is not dependent on the gradients of the model. We will include these points in the discussion in the final version.
>  - FreqRISE achieves higher performance in frequency domain for both the gender task and the digit task. When moving to the time-frequency domain, FreqRISE still achieves better performance on the digit task, but not on the gender task.
> We believe that the difference is to be found in the data properties. For the gender task, we believe that the frequency domain is sufficient for describing the data since the fundamental frequency is known to be discriminative of gender [1]. Thus, when moving to the time-frequency domain, we are adding unnecessary complexity and as a consequence, FreqRISE has lower performance.
> This is likely because FreqRISE performs best, when the salient information is sparse. On the other hand, for the digit task, we see in the example in the paper that frequency relevance is distributed non-uniformly across time. Using the time-frequency domain therefore adds additional insights into the model in this case. As of now, no existing methods are able to estimate the "correct" domain in which to explain a model. It is therefore up to the domain experts to choose which domain is more meaningful for them to look at.
> We have a brief discussion on this in the paper, but agree that it is insufficient. We will expand it to include the points mentioned here.
> ## Complexity
> We have added a discussion in the general reply, where we do a preliminary post-processing which enhances the complexity scores while keeping the other scores approximately the same. Additionally, we want to emphasize that while complexity is useful, faithfulness and localization are the only scores that actually says anything about the method’s ability to identify relevance. In our case, especially for the synthetic data and the digit task, FreqRISE substantially outperforms the baselines on these scores.
>
> Thus for some tasks, maybe additional complexity is needed to properly describe the data. Finally, we want to say that current methods for estimating complexity do not take smoothness into account. A relevance map with 10% of activated features randomly distributed across the map will therefore have the same complexity if the relevance was clustered around the same location. Future research should focus on quantifying some of these properties in addition.
> We hope these comments alleviate some of your concerns. We will add these points to the discussion.
>
> ## Superiority
> We believe this ties in with our points above. Given the increased performance in localization and faithfulness along with the potential reduction in complexity through thresholding, we believe FreqRISE is a promising new direction.
>
> Additionally, we want to emphasize that while LRP and IG both require access to the model gradients, FreqRISE is completely model-agnostic and can be applied to any model, even non deep learning models. This allows the users to gain critical insights into the model workings. More importantly, we present a new method in the highly under-developed field of explaining time series in more informative domains, such as frequency and time-frequency. We hope this research will pave the road for future development in this direction. We will update the discussion in the paper to reflect these points.
>
> ## Minor issues
> Thank you for pointing our attention to this. We have corrected the typos and updated the figure captions to include more details.
>
> We want to thank the reviewer for the great suggestions and hope our answers alleviated your concerns.
>
> [1] Calvert, D. R. (1988). CLINICAL MEASUREMENT OF SPEECH AND VOICE. Laryngoscope, 98(8), 905. https://doi.org/10.1288/00005537-198808000-00028

---

### Official Review · Reviewer_SgPr · 2024-10-09
**Interesting proposal, but lacking crucial information to establish its significance**

**Confidence:** 4

**Summary:**

The authors propose an explainable model to highlight the importance of frequencies in signals. By masking the time-frequency representation, the authors can output a relevance map of the frequencies by time, providing additional information compared to simply masking on either the time or frequency domains alone.

The proposal is tested against two baselines with two datasets (one synthetic and one real) for three tasks.

**Strengths:**

The paper presents an innovative approach by masking the time-frequency representation and producing a relevance map of the time-frequency representation.

**Weaknesses:**

The paper lacks details and misses important baselines. It needs rewriting to make the content clearer. Additionally, the story should be, in my opinion, revised from time series to signal as nothing guarantee that it can be applied to usual time series.

**Final Rebuttal Confidence:**

4

**Final Rebuttal Justification:**

Thank you to the authors for addressing my comments and engaging in an interesting exchange. I find the proposal compelling, and the results are promising. However, this version of the paper would benefit from additional comparative analysis and, above all, greater clarity to better establish the proposal's relevance. Additionally, the datasets used may not capture the full range of potential applications, making it challenging to generalize the performance findings presented.

Other reviewers also noted insufficient discussion of the results and a lack of guidance on how to use the proposal (especially select the different hyperparameters). Together, these points contribute to my recommendation to reject this paper in its current form.

**Justification:**

The proposal is interesting and the results look promising, but I suggest rejecting this paper as it lacks clarity and needs more comparison to clearly draw conclusions on its relevance. Please find below more details.

## Goal

According to the authors (line 111),
> Our aim is to explain the black box model $f(·)$.

Is this goal achieved? The authors manage to generate a time-frequency relevance map, but does it help in better understanding $f$? According to the experiments, it seems that the proposal increases complexity, so is this proposal more explainable?

Lines 323-325 state,
> [...] indicating that FreqRISE struggles more to identify relevant features in the time-frequency domain.

Doesn’t this mean that the proposal is not efficient?

## Clarity

### Figure 5

> The relevance map shows that most relevance is put just at the beginning of the signal.

As far as I understand the figure and following Figure 4, where dark blue shows relevance, the relevance in Figure 5 is around 0.5 (dark blue area), and not at the beginning, where it looks like there is no relevance. Important information is missing, such as the meaning of the color bar and why the time-frequency relevance is useful to explain $f$.

> This shows the benefit of having both the time and frequency component when computing relevance maps for the digit task.

But what can we conclude here? How is it relevant for the gender task? The frequency domain relevance seems sufficient. For the reader to really appreciate the importance of the time-frequency relevance, the authors should provide the equivalent of Figures 4 and 5 for the digit task and the synthetic dataset. As of now, it seems the time-frequency domain significantly increases complexity, improves results, but does not enhance explainability.

### Proposal

The proposal is freqRISE. Then, why do the authors sometimes refer to it as RISE or (freq)RISE? Something is not clear here. If I understand correctly, freqRISE is an enhanced version of RISE; it has the same properties as RISE (time-based relevance) and also produces time-frequency relevance. Therefore, when referring to freqRISE, the authors should use the name freqRISE, not RISE.

## Baselines

This brings me to my next comment: Why is RISE not a baseline? FreqRISE, being an enhanced version, should provide better performance than RISE, but this is not shown in the paper. If RISE only increases complexity without improving performance, does it make it a relevant enhancement? If the authors want to demonstrate that masking in the frequency domain is better than masking in the time domain, they need to compare with such solutions.

Additionally, why are TimeREISE and RELAX not baselines?

## Story
The proposal primarily focuses on signals rather than general time series data. Therefore, I would suggest replacing the mention of "time series" with "audio signals" or "time signals." There is no guarantee that the proposed methods and baselines would work for typical time series data such as electricity consumption, solar generation, traffic, weather, or exchange rates (especially for exchange rates, where frequency might not be representative). Time series data are usually collected over several years making datasets complex. Additionally, the authors mentioned that for audioMNIST, their proposal would require a large number of masks (20 000). For usual time series data, it would likely require even more masks (especially considering that some of these datasets have multiple features), significantly increasing complexity. For these reasons, I would revise the introduction to refer to "time signals" instead of "time series."



## Additional Comments for Future Revision

> [...] provide a comprehensive evaluation of our proposed approach across several datasets and tasks.

Only 2 datasets and 3 tasks are considered; can we talk about several?

Line 196:
> Both models achieve an accuracy of 100%.

This should be in the results discussion, not in the dataset description. The same applies to lines 215-217.

### Figure 1

Figure 1: STDFT in the caption should be the full name. But the framework in Figure 1 is a generic framework with $g$ being the function to transpose input from the time to frequency domain, so the caption should not mention STDFT or just that STDFT is an example of the function $g$.

### Figure 4

I am not sure what we are looking at in Figure 4, especially for LRP. More explanations would be required.  For FreqRISE, the blue line in the frequency domain are the relevant frequencies for the task, and then what do the blue lines in the time domain indicate? Are they the temporal positions where these frequencies are relevant? Or are they the relevant time steps.
LRP, working in the time domain, operates oppositely. It determines the relevant time step for the given task and then we can identify which frequencies correspond to these time steps in the frequency domain?

Also, the authors should use subcaptions such as a, b, c, d to make reading and referencing easier instead of using "top" and "bottom".

### Figure 3

The authors should compare their model with a dummy model that randomly selects the class. For instance, when determining the gender class, the dummy model has a 50% chance of being correct. A complex model that does not perform better than the dummy model should not be considered efficient. The dummy model, which purely guesses randomly, is not affected by removing frequencies and should have constant accuracy.

However, the relevance of the "baselines" **Rand.** and **Amp.** is unclear. It is not specified which model was used to obtain the corresponding plots. If the authors intend to compare different methods of removing frequencies (1. from most important to less important based on the relevance map, 2. randomly, and 3. based on amplitude), they should provide plots for each method using the different baselines (IG, LRP, freqRISE). Or at least provide more information of what **Rand.** and **Amp.** represents.

---

> ### Author Rebuttal · Authors · 2024-10-22
>
> We are grateful that the reviewer finds our approach innovative and interesting.
> We thank the reviewer for the review, as the questions raised will help clarify our paper. We have uploaded a revised version of the paper.
>
> # General rebuttal
> We include a general rebuttal here, since more reviewers asked about complexity scores.
> ## Updated faithfulness scores
> First of all, we want to mention that we found a small bug in how we had computed the faithfulness scores. We have updated the tables accordingly and uploaded a revised version of the paper. The absolute values have changed slightly, but none of them change any of the conclusions of the paper.
> ## Complexity
> The reviewers expressed concerns regarding the complexity of the relevance maps produced by FreqRISE. The reason for this is that, contrary to LRP and IG, FreqRISE is not able to assign 0 relevance to data points. This means that the baseline relevance will fluctuate on lower values, while FreqRISE will still assign high relevance to relevant points. We briefly mention that this might be solved through post processing. We did some preliminary experiments, where we set all values below a certain ratio of the maximum of the relevance map to zero.
>
> *Synthetic dataset*
> On the synthetic dataset we achieve **complexities of 0.65 and 0.64** in the low and high noise settings respectively, while still maintaining a **localization score of 100%** on both tasks.
>
> *AudioMNIST*
> Below, we show the complexity (C and faithfulness (F) results that we are able to obtain on AudioMNIST (prior and post thresholding).
>
> | Task   | Domain         | C. prior | C. post | F. prior | F. post |
> | ------ | -------------- | -------- | ------- | -------- | ------- |
> | Digit  | Frequency      | 8.17     | 7.01    | .160     | .182    |
> | Digit  | Time-frequency | 10.82    | 8.44    | .104     | .134    |
> | Gender | Frequency      | 8.01     | 7.33    | .416     | .417    |
> | Gender | Time-frequency | 10.78    | 8.69    | .423     | .422    |
>
> The results clearly indicate that we can achieve lower complexity through thresholding, while keeping competitive faithfulness scores.
> We also want to emphasize that here, the threshold is chosen across the dataset, while a smarter thresholding may improve results further.
> Finally, while complexity is a good measure for the quality of the relevance map it is not a complete picture. Low complexity with correspondingly low faithfulness or localization is not useful.
> We will extend the existing discussion in lines 383-389 to include the relevant points raised here. We will also include the preliminary post processing scores in the appendix.
>
> # Specific rebuttal
> ## Clarification
> The reviewer states: \
> “By masking the time-frequency representation, the authors can output a relevance map of the frequencies by time, providing additional information compared to simply masking on either the time or frequency domains alone.”\
> However, our proposed method FreqRISE masks either in the frequency domain or time-frequency domain (see lines 138-140). The chosen domain depends on the application. We are the first to present a method that produces explanations in the frequency or time-frequency domain by masking. I.e. our contribution is both masking in the frequency domain *and* in the time-frequency domain.
>
> We believe this also clarifies the question "Doesn’t this mean that the proposal is not efficient?" Lines 323-325 do not mean that the proposal is not efficient, but rather that for the gender task, the frequency domain is better suited for explanations. We agree that the argumentation can be clearer and we will update the discussion in the final version.
>
> ## Goal
> Thank you for your question. Previous work on time series explainability provides explanations as relevance maps over time [4, 5]. The recent work on virtual inspection layers [1] paved the road for future research on relevance maps in frequency and time-frequency domains. We develop a new method, FreqRISE, which uses masks to explain the models in these domains.
>
> As with all explainability methods, assessing their ability to explain models is difficult. However, we achieve much better localization scores on the synthetic dataset in the noisy setting and better results on faithfulness in the frequency domain for both AudioMNIST tasks and in the time-frequency domain for  the digit task. We have also included a discussion on the complexity scores in the general reply. These results lead us to conclude that we do a better job at explaining the signals in the frequency and time-frequency domains than current state-of-the-art methods.
>
> Additionally, we want to emphasize that while LRP and IG both require access to the model gradients, FreqRISE is completely model-agnostic and can be applied to any model, even non deep learning models. This allows the users to gain critical insights into the model workings.
> More importantly, we present a new method in the highly under-developed field of explaining time series in more informative domains. We hope this research will pave the road for future development in this direction.
> We will update the discussion in the paper to reflect these points.
>
> ## Clarity
> ### Figure 5
> Thank you for pointing this out, we have updated the text:
> > “The STDFT reveals that the signal starts around t=0.3s, where the relevance map puts most of the relevance. However, around t=0.5s, there are two distinct patterns emerging along the frequency axis. The fact that the method is allowed to distribute the relevance differently across frequencies depending on the time shows the benefit of having both the time and frequency component when computing relevance maps for the digit task.”
>
> Additionally, we have added a colorbar to the figure in the revised version. Regarding the gender task, we refer to the beginning of the rebuttal, where we argue that indeed the frequency domain is better suited for the gender task.
>
> We have included more examples of explanations in the appendix.
>
> ## Proposal
> The reviewer is indeed right that we are presenting FreqRISE - a novel masking-based framework for explaining time series in frequency and time-frequency domains. RISE is designed to produce explanations in the input domain (in this case, time). Therefore, when referring to our own method we use FreqRISE, and when producing explanations in the time domain we use RISE (see lines 215-217). We produce explanations in the time domain for all baselines (i.e. LRP and IG), as well as RISE, to allow the reader to compare against performances in frequency and time-frequency domains (our contribution).
>
> This leads us to our next point: Neither RISE, TimeREISE or RELAX are able to provide explanations in other domains than the input domain and it is therefore not possible to include them as baselines. Only one existing paper [1] has produced explanations in the frequency and time-frequency domain, and we include this as a baseline. This highlights that our approach is highly relevant as it, along with [1], introduces a new way of explaining time series.
>
> ## Story
> Thank you for your comments. For this work, we test our method on two different types of data: synthetic data and audio data. However, there are many other types of time series with information in the frequency domain, e.g. ECG [1], and EEG [2]. Additionally, recent work on building diffusion models in the frequency domain has shown the benefit in modeling the frequency content of various datasets spanning climate, finance and engineering applications [3]. Similarly for e.g. electricity consumption, we would often expect seasonal (i.e. frequency) trends in the data.
>
> This being said, we agree that the user should always consider what is meaningful for the dataset at hand. This includes considering which domain to explain the signal in. We provide a new tool for providing explanations in the frequency domain and time-frequency domain for time series, which will generally work for all uniformly sampled time series. We will update our discussion to reflect these points.
>
> ## Additional comments
> We have changed the wording from "several” to “two datasets and four tasks” (we have both a low and high noise setting for the synthetic dataset). \
> We have moved the accuracies of the models to section 4.1 and 4.2.
>
> ### Figure 1
> We have updated the description of Figure 1, please see the revised paper.
>
> ### Figure 4
> We use RISE and LRP to compute relevance maps in the time domain as well to show that the relevance is more localized in the frequency domain. Additionally, we have added colorbars to the plots.
> We hope this clarifies your question.
>
> ### Figure 3
> We adhere to the evaluation presented in [1] and evaluate all scores on the trained black box model, f, which is considered frozen for all experiments.
> All baselines are included in Figure 3. We state that: “We have also included two additional baselines, namely randomly deleting features and deleting features based on their amplitude.” We agree that this is unclear and have updated the description to:
> > “We have also included two additional baselines, namely randomly deleting features (**Rand.**) and deleting features based on their amplitude (**Amp.**).”
>
> Thank you for all your comments and suggestions.
>
> [1] Vielhaben, J. et al (2024). Explainable AI for time series via Virtual Inspection Layers. Pattern Recognition.\
> [2] Bucci, P. et al (2011). Normal EEG Patterns and Waveforms. Standard Electroencephalography in Clinical Psychiatry: a Practical Handbook, 33–57.\
> [3] Crabbé, J. et al (2024). Time Series Diffusion in the Frequency Domain. Proceedings of Machine Learning Research.\
> [4] Crabbé, J., & van der Schaar, M. (2021). Explaining Time Series Predictions with Dynamic Masks. Proceedings of Machine Learning Research.\
> [5] Liu, Z. et al (2024). TIMEX++: Learning Time-Series Explanations with Information Bottleneck. Proceedings of Machine Learning Research.

---

### Meta-Review · Area_Chair_kTGG · 2024-11-01

**Recommendation:** Accept (Poster)
**Confidence:** 3

**Metareview:**

The paper presents a new way to do XAI in the frequency domain of models which classify time-series.

The idea is potentially novel but the presentation is significantly unclear.
The basic definition given in equation 3 is mathematically quite unclear,
- even if one tries to match this against the RISE paper, https://arxiv.org/pdf/1806.07421

Given the constraints of time it is quite challenging to ascertain the correctness with confidence,
but I tend to believe that with a significant overhaul of the writing maybe the ambiguities can be resolved.

**Suggested Changes To The Recommendation:**

1: I agree that the recommendation could be moved down

---

### Decision · Program_Chairs · 2024-11-06

**Decision:**

Accept (Poster)

**Comment:**

We recommend a poster presentation given the AC and reviewers recommendations.